# Fertility-Sparing Treatment of Patients with Endometrial Cancer: A Review of the Literature

**DOI:** 10.3390/jcm10204784

**Published:** 2021-10-19

**Authors:** Umberto Leone Roberti Maggiore, Rola Khamisy-Farah, Nicola Luigi Bragazzi, Giorgio Bogani, Fabio Martinelli, Salvatore Lopez, Valentina Chiappa, Mauro Signorelli, Antonino Ditto, Francesco Raspagliesi

**Affiliations:** 1Gynecologic Oncology, Fondazione IRCCS Istituto Nazionale Tumori, 20133 Milan, Italy; giorgio.bogani@istitutotumori.mi.it (G.B.); fabio.martinelli@istitutotumori.mi.it (F.M.); salvatore.lopez@istitutotumori.mi.it (S.L.); valentina.chiappa@istitutotumori.mi.it (V.C.); mauro.signorelli@istitutotumori.mi.it (M.S.); antonino.ditto@istitutotumori.mi.it (A.D.); raspagliesi@istitutotumori.mi.it (F.R.); 2Clalit Health Service, Akko, Azrieli Faculty of Medicine, Bar-Ilan University, Safed 13100, Israel; rkhamisy@yahoo.com; 3Laboratory for Industrial and Applied Mathematics (LIAM), Department of Mathematics and Statistics, York University, Toronto, ON M3J 1P3, Canada; dottornicolaluigibragazzi@gmail.com

**Keywords:** endometrial cancer, fertility-sparing, hysteroscopy, metformin, progestin

## Abstract

Endometrial cancer (EC) is currently the most common malignancy of the female genital tract in developed countries. Although it is more common in postmenopausal women, it may affect up to 25% in the premenopausal age and 3–5% under the age of 40 years. Furthermore, in the last decades a significant shift to pregnancy at older maternal ages, particularly in resource-rich countries, has been observed. Therefore, in this scenario fertility-sparing alternatives should be discussed with patients affected by EC. This study summarizes available literature on fertility-sparing management of patients affected by EC, focusing on the oncologic and reproductive outcomes. A systematic computerized search of the literature was performed in two electronic databases (PubMed and MEDLINE) in order to identify relevant articles to be included for the purpose of this systematic review. On the basis of available evidence, fertility-sparing alternatives are oral progestins alone or in combination with other drugs, levonorgestrel intrauterine system and hysteroscopic resection in association with progestin therapies. These strategies seem feasible and safe for young patients with G1 endometrioid EC limited to the endometrium. However, there is a lack of high-quality evidence on the efficacy and safety of fertility-sparing treatments and future well-designed studies are required.

## 1. Introduction

Endometrial cancer (EC) is currently the most common malignancy of the female genital tract in developed countries; in Europe, EC has shown a 5-year prevalence of 34.7% (445,805 cases) [1]. In 2018, the estimated number of new EC cases in Europe was 121,578 with 29,638 deaths, with aging and increasing obesity among women representing the two principal risk factors [2]. EC is more common among patients of postmenopausal age, but about 25% women are premenopausal and 3–5% are younger than 40 years [3].

Hysterectomy with bilateral salpingo-oophorectomy plus nodal evaluation and with or without peritoneal staging represents the standard therapy of EC [4,5,6]. Although radical surgery is associated with 5-year oncologic survival outcome of 75–90% of patients [7], it prevents the possibility to have future pregnancies [8,9].

Thus, the standard surgical treatment may not be suitable for patients wishing to maintain their fertility. Therefore, fertility-sparing alternatives should be thoroughly explained to EC women, discussing the oncologic outcomes related to each approach. Fertility-sparing treatments can be proposed to patients with endometrioid intra-epithelial neoplasia (EIN) or grade 1 EC without myometrial invasion [2]. Different conservative modalities have been demonstrated safe and feasible such as oral/local progestin treatment +/− hysteroscopic resection of endometrial lesions [10].

The aim of this review is to summarize available evidence on fertility-sparing options for patients affected by EC, focusing on the oncologic and reproductive outcomes.

## 2. Material and Methods

A systematic review of the available evidence, from 1950 until December 2021 (last research 1st May), was conducted consulting two electronic databases (PubMed and MEDLINE) to select relevant articles. All relevant papers were evaluated, and references were evaluated to identify other articles for potential inclusion in the current review. All articles were assessed by two independent reviewers (U.L.R.M. and G.B.) and in the case of discrepancy a third author (F.R.) was asked to participate for consensus. Firstly, eligibility was evaluated considering the titles and abstracts. Full manuscripts were obtained for all selected studies and decision for final inclusion was made after detailed examination of the papers. Randomized controlled trials (RCTs), prospective studies, case–control studies, and retrospective cohort studies were considered for inclusion in the review.

Two independent authors (U.L.R.M., G.B.) ran a specific literature search. The search included the term endometrial cancer in combination with other keywords and medical subjects heading terms such as fertility preservation, fertility-sparing, hysteroscopic resection, levonorgestrel, oral progestin. Given the aim of this narrative review, which focuses on different fertility-sparing methods for the treatment of EC, we arbitrarily decided not to use a systematic approach in reporting results; thus, we reported more relevant studies for each kind of treatment to provide the reader a complete and concise overview of the available evidence on the fertility-sparing management for EC.

## 3. Results

Several types of treatment have been described for the fertility-sparing management of EC; however, no consensus is established on which agent, dose, or duration of treatment is more efficacious. Progestins, metformin, and hysteroscopic resection are the most common investigated modalities to conservatively treat EC (Table 1 and Table 2).

### 3.1. Oral Progestins

Traditionally, the most frequently prescribed drugs for conservative EC treatment are medroxyprogesterone acetate (MPA) or megestrol acetate (MA) and several studies have demonstrated their efficacy and safety [16,17,18,29,30,35,36,37,38,39,40,41]. The duration of progestin therapy and the type of progestin and dosages administered in different studies are heterogeneous. MPA has been used mainly continuously with doses ranging between 20 and 1500 mg/day, and MA was used at 40–480 mg/day. The administrations of MPA at 400–600 mg/day or MA at 160–320 mg/day are those more frequently reported and, therefore, suggested [14]. A Japanese multicenter prospective study was carried out including 28 female patients with stage IA EC and 17 with atypical endometrial hyperplasia (AEH). They were treated with a daily MPA (600 mg) plus low-dose aspirin for a total of 26 weeks. Endometrial biopsies were performed after 8 and 16 weeks of therapy. In total, 55% of women with EC and 82% of women with AEH had a CR, with an overall CR rate of 67%. At follow-up, 12 pregnancies and seven normal deliveries were observed after treatment. Fourteen (47%) recurrences were recorded between 7 and 36 months [17]. A single center prospective study evaluated the efficacy of daily 160 mg MA (initial dose) for conservative treatment of 21 patients with stage IA G1 EC over a 6-month period. Eighteen patients (85.7%) had a CR and 3 women had radical surgery. CR was found in 5 women (27.8%) with a dose of 160 mg/day, while 13 women (72.2%) responded with 320 mg/day. Pregnancy occurred in 5 patients (27.8%). Three of 18 (16.7%) patients had a recurrence [16]. Although most of the papers have shown that median time interval to obtain CR is about 6 months [14], two studies suggest that the response rate seems to raise with the length of treatment, achieving a plateau at 12 months [11,12]. Koskas and coauthors demonstrated a CR rate after 3, 6, 12, 18, and 24 months of treatment of 30.4%, 72.4%, 78.0%, 80%, and 81.4%, respectively [11]. A recent paper by Cho and coworkers evaluated the efficacy of progestin treatment in women who had not CR after 9 months of therapy. Fifty-one patients with stage IA, G1/2 endometrioid EC who with persistence at endometrial obtained at 9–12 months after at least 9 months of progestin-based therapy were included in this study. CR after prolonged progestin treatment was recorded among 37 women (72.5%). Median time to CR from the beginning of treatment was 17.3 months (range, 12.1–91.7 months). Patients who did not have PR until 12 months were at higher risk of failure to CR after prolonged medical therapy (OR, 21.803, 95% CI, 3.601–132.025, *p* = 0.001) [13].

### 3.2. Levonorgestrel Intrauterine System

An alternative to oral progestins is represented by levonorgestrel intrauterine system (LNG-IUS). This device acts through the local release of the second-generation progestin levonorgestrel, thus combining two main advantages: the reduction of systemic adverse effects and the increase of local effectiveness causing endometrial decidualization and atrophy [42]. Different studies have evaluated the efficacy of LNG-IUS for the fertility-sparing management of EC [20,43,44,45,46]. In our institution, a retrospective study was performed to investigate the effectiveness of LNG-IUS treatment in women affected by AEH or EC. Forty-eight patients were included in the study, among them 28/48 had AEH, 16/48 had G1 EC, and 4/48 had G2 EC. Women with G1 EC, 13/16 (81.3%) had a CR while 3/4 (75.0%) patients with G2 EC had a CR with a mean (SD) time to CR of 5.0 ± 2.9 months and 4.0 ± 0 months, respectively. Eight out of 16 (50.0%) patients with G1 EC attempted to conceive while no patient with G2 EC actively tried to achieve a pregnancy. All patients had a pregnancy which was obtained through assisted reproductive techniques (ART) in 6/8 (75.0%) cases [20]. Comparable results have been presented by Pal et al. in another retrospective study including 46 patients with AEH or early-grade EC (15 (47%) had AEH, 9 (28%) had G1 EC and 8 (25%) had G2 EC) treated with LNG-IUS. Overall response rate was 75% (95% CI 57–89) at 6 months, 67% (95% CI 30–93) in G1 EC and 75% (CI 35–97) in G2 EC. Interestingly, non-responder patients had a bigger uterine size measured by uterine largest diameter (9.3 versus 8 cm). No information about reproductive outcomes was reported in this study [46]. LNG-IUS has been tested also in combination with other medications [19,21,32,47,48,49]. In 2019, a Korean prospective multicenter study was conducted including 44 women with G1 EC confined to the endometrium and treated with combined oral MPA (500 mg/day)/LNG-IUS. At six months, CR rate was 37.1% (13/35). PR was observed in 25.7% (9/35) of patients. Progressive disease and treatment-related complications were not reported [49]. A retrospective study, including 118 patients with stage Ia G1/G2 EC receiving combined oral MPA (500 mg/day)/LNG-IUS, assessed oncologic and reproductive outcomes. Seventy-one (60.2%) patients had CR, and 49 of these patients (69.0%) attempted to conceive. Twenty-two (44.9%) patients had a pregnancy (30 pregnancies were recorded) [32]. Pronin and colleagues performed a prospective study enrolling 70 women aged less than 42 years with a diagnosis of AEH or G1 EC. Patients with AEH received monotherapy with levonorgestrel-releasing intrauterine system. Patients with G1 EC were treated with LNG-IUS combined with gonadotropin-releasing hormone agonist (subcutaneous injection of 3.6 mg gosereline acetate given every 28 days). All women used a hormonal treatment for at least 6 months. CR was reported in 23 (72%) women with EC and 35 (92%) with AEH. At follow-up, 2 of these responding patients with EC and 1 with AEH experienced a recurrence. Nine women (7 with EC and 2 with AEH) had persistent disease. Ten conceptions were achieved by 8 women, with 8 live births [21].

### 3.3. Metformin

Metformin is an insulin sensitizer since it enhances signaling through the insulin receptor, leading to an improvement in insulin resistance, followed by a decrease in circulating insulin levels. Furthermore, evidence indicates that metformin’s key target of action is the inhibition of hepatic gluconeogenesis, causing a secondary decline in insulin levels. Interestingly, Cantrell et al. demonstrated that metformin is a potent inhibitor of cell proliferation in EC cell lines through AMPK activation and subsequent inhibition of the mTOR pathway, paving the way for its potential use for EC prevention and treatment [50]. Subsequent studies showed that metformin may also promote progesterone receptor expression [51], exert anti-invasive and antimetastatic effects in human EC cells [52], and reverse progestin resistance in EC cells [53]. A phase 2 study including 17 patients with AEH and 19 patients with stage IA EC evaluated the effectiveness of metformin to decrease recurrence after treatment with MPA. Thirty-six patients received MPA 400 mg/day, low dose aspirin, and metformin 750 mg/day. Metformin dosages were progressively increased up to 2250 mg/day or the highest tolerated dose. After remission, metformin was extended until conception or disease recurrence, and patients received low dose estroprogestins or progestin for 6 cycles. Progression occurred in two women (6%) at 12-week follow-up. At 9-month follow-up, 29 women (81%) had a CR, and 5 (14%) patients had a PR. At follow-up, recurrence was observed in three women with previous CR (relapse rate, 10%). A relapse-free survival (RFS) rate of 89% was reported at 3-year follow-up. Severe toxicities were not reported [24]. In 2019, the same group of authors conducted a retrospective study assessing long-term outcomes of MPA (400 mg/day) plus metformin (750–2250 mg/day) patients with AEH (21 women) or EC (42 women). Metformin was given until conception, also after MPA interruption. CR was shown in 69 women (97%) at 18 months; CR rates observed at 6, 8–9, and 12 months were 60%, 84%, and 90%, respectively. Eight of 61 women (13.1%) had a recurrence after initial CR, with median follow-up of 57 months (13–115 months). Five-year RFS was 84.8% [26]. A recent retrospective cohort study compared progestin treatment plus metformin (Prog-Met) to progestin alone (Prog) for conservative management of patients with AEH/EIN or EC limited to the endometrium. Progestin used in this study were the following: MA at 80 to 160 mg orally daily, MPA at 10 to 40 mg orally daily, progesterone 400 mg orally daily, or LNG-IUD at 52 mg. The indication and duration for metformin 500–1000 mg daily was obtained from medical records. Ninety-two women were enrolled in this study, 54 (59%) were had AEH/EIN and 38 (41%) had EC. Progesterone alone was administered to 58 women (63%) while 34 (37%) received progesterone plus metformin. Overall, a response to treatment was demonstrated by 73 women (79%) while CR was achieved by 63 patients (69%). Similar CR rates or time to CR were observed between the two study groups. Disease recurrence occurred among 22% of patients. A total of 16 pregnancies (17%) was reported, all resulting in live births. Patients treated by progestin only therapy had a higher rate of pregnancy/live birth (24%) compared to those treated by progestin plus metformin. Out of 16 pregnancies, 13 (81%) were obtained by ART [23]. Recently, a randomized, open, blinded-endpoint design phase IIb dose response trial (FELICIA trial) has been announced. The main objective of this study is to elucidate the appropriate dose of metformin to be associated with MPA therapy for conservative management of patients with AEH and EC. Three-year RFS rate is the primary aim of the trial. The secondary objectives are the overall rate of response to MPA therapy, the conception rate after therapy, pregnancy outcomes, safety and toxicity profile, and modifications in insulin resistance and BMI. Fifteen Japanese institutions plan to enroll patients for an estimated ì sample size of 120 women within a 2.5-year period with a total follow-up period of at least 3 years [54].

### 3.4. Hysteroscopic Resection plus Progestin Therapies

An alternative EC fertility-sparing treatment is represented by the combination of hysteroscopic resection followed by progestin therapies. In 2010, a prospective study described for the first time a new technique to maintain fertility in 6 patients with early-stage IA EC with use of hysteroscopic resection combined with 160 mg of MA. This method consisted in a three-step procedure with a pathologic evaluation at every step: the resection of the disease (step 1), the resection of the endometrium next to the disease (step 2), and the resection of the myometrium underlying the disease (step 3). This fertility-sparing surgical technique demonstrated efficacious since both transvaginal ultrasound assessment and diagnostic hysteroscopy at 3, 6, 9, and 12 months after first surgical evaluation were negative for atypia or malignancy. Furthermore, 4/6 women (66%) had live births [33]. In 2011, another study aimed to verify the outcomes of combined operative hysteroscopy plus progestin as conservative treatment of young patients with FIGO Stage IA EC. Fourteen women wishing to preserve fertility were enrolled in this study and treated by hysteroscopic resection of the tumor and the underlying myometrial, combined with oral MA 160 mg/day for 6 months (6 women) or 52 mg LNG-IUS for 12 months (8 women). At a median follow-up of 40 months (range 13–79 months), one woman had a relapse after 5 months from surgery and underwent hysterectomy, whereas one woman was found with an endometrial hyperplasia without atypia at the 3- and 6-month hysteroscopic follow-up, with subsequent negative follow-ups. Three women attempted to achieve a pregnancy and one had a live birth [55]. A prospective study by the same institution reported their 15-year institutional experience of conservative management of EC patients by using a combination of hysteroscopic resection and medical therapies (oral MA or LNG-IUS). A total of 28 women with FIGO stage IA, G1, and 2 endometrioid EC, aiming to maintain fertility were included in this prospective trial. At 3 months, 25 women (89.3%) demonstrated a CR, two (7.1%) had persistent disease, whereas one (3.6%) with progressive disease underwent definitive surgery and final pathologic examination showed a FIGO stage IA, G3 endometrioid EC. At 6 months follow-up, one woman with persistent disease had radical surgery (stage IA, G1 endometrioid), while the other one was successfully re-treated. Two cases of relapse were reported (7.7%) and in both of the cases EC and synchronous ovarian cancer were observed. CR lasted for a median of 94.5 months (range, 8-175 months). In most of the cases women who responded (57.7%) tried to have a pregnancy (93.3% and 86.6% pregnancy and live birth rates, respectively) [28]. Yang et al. published the largest study evaluating the effectiveness of hysteroscopic assessment and lesion surgical ablation plus medical treatment in women with endometrial AEH and early-stage EC. Women with AEH (*n* = 120) or G1 EC (*n* = 40, FIGO stage IA) were retrospectively enrolled in this study. All women were administered continuous oral progestin associated with hysteroscopic biopsy every 3 months until CR. Overall, 148 women (97.4%) had CR while 3 AEH and 1 EEC patients had a disease progression. The mean time to CR was 6.7 ± 0.3 months (range, 1–18 months). Among 60 women who tried to obtain a pregnancy after achieving CR, 45.0% (27/60) succeeded, 25.0% (15/60) had a live birth, 13.3% (8/60) were still in pregnancy, while 6.7% had a spontaneous miscarriage [27]. Recently, Mazzon et al. published the long-term follow-up of 6 patients who underwent hysteroscopic resection plus MA and achieving CR showing that after a median time of 16 years all patients had no disease relapse [56].

### 3.5. Comparison among Different Treatment Options

In 2017, two systematic reviews and meta-analyses on the outcomes of various fertility-sparing treatments for EC were published [15,22]. Fan et al. aimed to evaluate the effectiveness of different therapies for grade 1 presumed stage IA EC. A total of 28 studies including 619 patients were considered for this review. Patients who were treated only by oral progestin (456 women) had a CR, recurrence rate (ReR), and pregnancy rate (PregR) of 76.3%, (95% confidence interval (CI), 70.7–81.1%); 30.7% (95% CI, 21.0–42.4%); and 52.1% (95% CI, 41.2–66.0%), respectively. Ninety women using LNG-IUS had a CR, ReR, and PregR of 72.9% (95% CI, 60.4–82.5%); 11.0% (95% CI, 5.1–22.0%); and 56.0% (95% CI, 37.3–73.1%), respectively. The group of patients treated by hysteroscopic resection combined with use of progestins (73 women) had a CR, ReR, and PregR of 95.3% (95% CI, 87.8–100%); 14.1% (95% CI, 7.1–26.1%); and 47.8% (95% CI, 33.0–69.5%), respectively [22]. Zhang et al. aimed to evaluate disease regression, recurrence, and live birth rate among patients with G1 early-stage EC and AEH treated with conservative therapies including oral progestins, hysteroscopic resection, and the LNG-IUS. This systematic review and meta-analysis finally included 54 studies. This study demonstrated that hysteroscopic resection followed by oral progestins versus oral progestins alone caused a higher pooled regression (98.1% vs. 77.2%) and live birth rate (52.6% vs. 33.4%) and a lower recurrence rate compared with (4.8% vs. 32.2%). Similarly, hysteroscopic resection plus oral progestins showed a significant higher pooled live birth rate (52.6% vs. 18.1%) than LNG-IUS alone. No significant differences in regression (98.1% vs. 94.2%) and relapse rates (4.8% vs. 3.9%) were recorded [15].

## 4. Discussion

Over the last years, there has been a significant shift to pregnancy at older maternal ages, particularly in resource-rich countries [57]. In the USA, pregnancy rates have decreased for women under 30 years and raised for women age 30 and above from 1990 to 2015 [58]. Therefore, according to this scenario, it is crucial that fertile women with a diagnosis of gynecological malignancy are offered an oncofertility service to maximize the reproductive potential and to counsel about fertility preservation options of cancer patients and survivors. Fertility-sparing treatment alternatives have been proposed for the three major gynecologic cancers: cervical, ovarian, and endometrial cancer. Overall, fertility-sparing management of gynecologic cancers is associated with acceptable rates of PFS and OS [59].

The association between epidemiological risk factors and the progression to EC can be justified by the unopposed estrogen hypothesis [60]. In this relationship, it has been shown that progesterone may reverse this neoplastic process, by opposing the action of estrogen on the endometrium. In detail, treatment with progesterone/progestin may inhibit estrogen receptors, inducing a decrease in endometrial cell mitosis, promotion of apoptosis, and production of secretory endometrium. The use of progesterone/progestins to treat endometrial hyperplasia and cancer has been observed for decades [61,62,63,64]. On the basis of available studies, it has been demonstrated that the risk of persistence or progression of endometrial hyperplasia in women using progestin therapies is about 1% for simple hyperplasia, 3% for non-atypical complex hyperplasia, and 15–75% for ACH [65]. Either oral or local progestins, alone or in association with other drugs (metformin) or treatment (hysteroscopic resection), have been studied for conservative management of patients with ACH/EC [15,22,66,67]. Available data on conservative treatment of EC patients are based on small retrospective or prospective observational studies. Head-to-head comparison trials are almost unavailable, various regimens are described in terms of drugs, dosages, length of treatment, and follow-up. Therefore, the optimal fertility-sparing management for EC is still matter of research. According to the most recent ESGO/ESTRO/ESP guidelines for the management of patients with EC, MPA (400–600 mg/day) or MA (160–320 mg/day) is the recommended treatment due to the largest number of published data. Treatment with LNG-IUS can also be prescribed as well as hysteroscopic resection before starting medical therapies that may offer patients an additional benefit in terms of outcome [2].

Patients wishing to preserve fertility should be referred to tertiary centers. Transvaginal ultrasound performed by an expert sonographer can be used as alternative to MRI [68]. Selection of ideal candidates is a fundamental point and should identify those patients the lowest risk of metastatic cancer or local invasion and thus the highest likelihood of CR. Therefore, the ideal patients for conservative management are represented by young patients with well-differentiated endometrioid EC limited to the endometrium. Among these women, endometrial sampling, ideally by hysteroscopy, should be performed and the histologic diagnosis must be posed/confirmed by a pathologist specifically trained in gynecological pathology [2]. Scanty evidence is available on the oncologic outcomes of patients with G2–G3 disease. A recent Gynecologic Cancer Inter-Group study aimed to report the oncological and reproductive outcomes of 23 patients affected by G2 endometrioid EC limited to the endometrium showing that conservative treatment seemed to also be efficacious in these types of tumors. However, the potential pathological undegrading or non-endometrioid histology misdiagnosis should be taken into consideration [69]. Furthermore, Park et al. demonstrated that the use of progestins seems to be feasible in women affected by stage IA, G2-3 differentiation limited to the endometrium and those affected by stage IA G1 differentiation with superficial myometrial invasion [37]. However, the paucity of high-quality studies on this issue does not support the widening of the criteria for target patients of EC fertility-sparing treatment, so far. In addition, Casadio and colleagues have shown a proof of concept that conservative treatment may also be considered among patients with initial myometrial infiltration. However, these promising findings should be confirmed with future randomized and multicentric studies [70,71]. Finally, the classification of EC was revolutionized in 2013 with the identification of four molecular subtypes of EC, based on genomic architecture, by The Cancer Genome Atlas (TCGA) Research Network [72]. Recently, a retrospective study including 57 patients aimed to evaluate the prognostic significance of the molecular classification in the fertility-sparing management of EC. This study demonstrated that patients with mismatch repair deficiency had a significantly lower CR or PR rate than those with wild-type p53 in terms of the best overall response and CR rate at 6 months [73]. Therefore, we deem that future studies investigating conservative management of EC should also include the molecular classification of EC, since it may represent a crucial biomarker to plan treatment and counsel the patient.

Despite the effectiveness of different fertility-sparing strategies for EC, some patients do not respond to treatment or they may recur after an initial regression of the disease, demonstrating a risk of progression to invasive cancer. For this reason, great efforts have been spent to identify predictive factors of response to conservative treatment including trials on clinical, pathological, and immunohistochemical characteristics [74,75,76], particularly on the role of estrogen receptor and progesterone receptor, whose expression is easily assessable by immunohistochemistry. Raffone and colleagues have shown in their meta-analysis on this topic that progesterone receptor expression was related to the response of AEH and EC in patients treated by LNG-IUS. However, they showed that the predictive accuracy was not reliable to be of clinical utility as a stand-alone marker [77]. More promising findings have been observed from the study of isoform B of the progesterone receptor. Interestingly, the same group of authors have recently shown that a low stromal isoform B progesterone receptor expression may represent a highly sensitive predictive marker in patients with AEH and/or EEC without response or in those who relapse conservatively treated with hysteroscopic resection followed by LNG-IUS insertion [78]. This observation may help clinician to select the ideal patients for fertility-sparing treatment of EC and combining PRB with other markers may permit the development of more accurate predictive models to optimize the treatment of these women.

The assessment of the response is fundamental, but no universally shared standard protocol has been currently developed. Different follow-up timepoints have been described, the most common being 3 months [79]. Endometrial post-treatment response may be evaluated with dilation and curettage, endometrial aspiration biopsy, or hysteroscopic biopsy. According to ESGO/ESTRO/ESP guidelines, to evaluate the degree of response to treatment, hysteroscopic guided biopsy and imaging at 3–4 and 6 months should be ruled out. If no response is achieved after 6 months, standard surgical treatment is recommended. Indeed, patients should be carefully counselled on conservative treatment and they should be informed it is not a standard management and offers a time frame for these women to attempt to conceive. Only women who strongly wish to maintain fertility are candidates for this strategy. Patients should accept close and regular visits and be informed of the need for radical surgery in case of no response to treatment and/or after pregnancies [2]. Most commonly, CR is achieved between 3 and 6 months from the beginning of fertility-sparing treatment. Notably, no consensus yet exists on the opportunity of a maintenance treatment. However, it seems reasonable to continue hormonal treatment in responders who wish to delay pregnancy. Therefore, patients who follow a fertility sparing treatment should be actively informed and encouraged to try for a pregnancy as soon as possible. Positive factors for successful pregnancies are represented by normal BMI (<24), a shorter time to CR, a prolonged three-month treatment, fewer hysteroscopy procedures, and a thicker endometrium, whereas recurrence before pregnancy may have a negative effect on conception [80]. Patients wishing to conceive can choose between natural methods and ART immediately without waiting. According to Fan et al., the pooled pregnancy rate was 75.3% after assisted reproductive techniques and only 39.3% in the group who adopted natural approach [22]. In addition, a recent study demonstrated that no significant difference was detected in terms of cumulative recurrence free survival when compared between ART cases and those involving natural conception [80].

Finally, treatment failure and patient recurrence should be considered. In the first case, ESGO/ESTRO/ESP guidelines state that if no response is achieved after 6 months of fertility-sparing therapy, standard surgical treatment is recommended [2]. Similarly, patients who experience recurrence after initial response should be counseled for radical surgery. However, some authors have proposed retreatment with progestins in this population of patients [81,82,83]. In these studies, CR was observed in very high percentage of women (>90%), however patients who underwent second-line fertility-sparing therapy experienced a worse recurrence rate with lower 5-years recurrence-free survival, despite a similar pregnancy rate [81]. According to ESGO/ESTRO/ESP guidelines fertility-sparing treatment can be considered for intrauterine recurrences only in highly selected cases under strict surveillance [2].

## 5. Conclusions

On the basis of available evidence, fertility-sparing strategies seem feasible and safe for young patients with G1 endometrioid EC limited to the endometrium. However, there is a lack of high-quality evidence on the efficacy and safety of fertility-sparing treatments and future well-designed studies are needed to offer stronger evidence on this issue. Furthermore, it is of primary importance that future studies on this topic should also include the molecular classification of endometrial cancer in order to enable early stratification and risk assignment to direct care.

Selected and strongly motivated women should be carefully counseled about the non-standard nature of fertility-sparing strategies and only once they have fully understood the potential risks of this management should they start conservative treatment.

## Figures and Tables

**Table 1 jcm-10-04784-t001:** Efficacy outcomes of fertility-sparing therapies.

	Dose	Time to CR	Patients Achieving CR	Recurrence Rate
Oral progestins: - MA - MPA	40–480 mg/day 20/1500 mg/day	Median at 4.5–6.3 months [11,12,13,14] Plateau at 12–18 months [11,12,13,14]	55.0–85.7% [15,16,17,18]	16.7–46.6% [16,17]
LNG-IUS (alone or plus oral progestins)	20 mcg/day	Mean: from 5.0 ± 2.9 to 9.8 ± 8.9 months [19,20]	72.9–87.5% [19,20,21]	11.0% [22]
Metformin (plus oral progestins)	750–2250 mg/day	Median at 5.9 months [23]	80.0% [24,25]	10.0–13.1% [24,26]
Hysteroscopic Resection (plus oral progestins or LNG-IUS)	-	Mean: 6.7 ± 0.3 months [27]	89.3–97.4% [27,28]	14.1% [22]

CR: complete response; LNG-IUS: levonorgestrel system; MA: megestrol acetate; MPA: medroxyprogesterone acetate.

**Table 2 jcm-10-04784-t002:** Fertility outcomes after fertility-sparing therapies.

	Dose	Percentage of Patients Trying to Conceive	Live Birth Rates
Oral progestins: - MA - MPA	40–480 mg/day 20/1500 mg/day	60.7–100% [16,17,29,30,31]	32.3% (95% CI, 22.9–42.5) [15]
LNG-IUS (alone or plus oral progestins)	20 mcg/day	50.0–69.0% [20,21,32]	18.1% (95% CI, 7.4–32.1) * [15]
Metformin (plus oral progestins)	750–2250 mg/day	NA	21.6% [14]
Hysteroscopic Resection (plus oral progestins or LNG-IUS)	-	21.4–83.3% [28,33,34]	52.6% (95% CI, 24.7–79.6%) * [15]

LNG-IUS: levonorgestrel system; MA: megestrol acetate; MPA: medroxyprogesterone acetate; NA: not available, * also including atypical endometrial hyperplasia.

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
