# Peer review of "Fertility-Sparing Treatment of Patients with Endometrial Cancer: A Review of the Literature"

_jcm, 2021, doi:10.3390/jcm10204784_

Round 1

Reviewer 1 Report

Fertility-sparing treatment of patients with endometrial cancer: a review of the literature is an interesting article that gives an overview of the current literature about this topic.  The article is well written and the subject matter is focused. The request revisions were well developed.

Author Response

We thank Reviewer 1 for the positive comment on our revised manuscript.

Reviewer 2 Report

This review accurately summarizes data available in the literature on possible fertility-sparing alternatives in patients with endometrial cancer. It is clear and well written.

Correct Autohors in the title.

LINE 314  treatament

Author Response

We than Reviewer 2 for the positive comment on our manuscript.

We apologise for the inaccuracies and we corrected them in the revised manuscript.

Reviewer 3 Report

This article is a review of fertility-sparing treatment of patients with endometrial cancer. It is comprehensive and provide a useful information. However, several points shown below are required to be answered.

  1. The authors did not mention the risk of pregnancy outcome after successful fertility-sparing treatment for endometrial cancer, compared with normal pregnancy. Please comment on this topic.
  2. There are several minor grammatical errors in the article. Please check the article carefully.
  3. In line 126, “Interestingly, non-responder patients had a bigger uterine uterine size measured by uter-126 ine largest diameter (9.3 versus 8 cm).” The word “uterine” is repeated. Is this correct?
  4. In line 207, “figo Stage IA EC” should be “FIGO stage IA EC”.
  5. In line 235, “45.0% (15/60) succeeded” appeared. Is this correct?

Author Response

This article is a review of fertility-sparing treatment of patients with endometrial cancer. It is comprehensive and provide a useful information. However, several points shown below are required to be answered.

1. The authors did not mention the risk of pregnancy outcome after successful fertility-sparing  treatment for endometrial cancer, compared with normal pregnancy. Please comment on this topic.

We thank the Reviewer for this comment. Thus, we added this to the Discussion of the revised manuscript (lines 387-390).

2. There are several minor grammatical errors in the article. Please check the article carefully.

We apologise for these inaccuracies and we corrected them throughout the whole revised manuscript.

3. In line 126, “Interestingly, non-responder patients had a bigger uterine uterine size measured by uterine largest diameter (9.3 versus 8 cm).” The word “uterine” is repeated. Is this correct?

We thank the Reviewer for this comment. This was a mistake and we fixed it.

4. In line 207, “figo Stage IA EC” should be “FIGO stage IA EC”.

We corrected this inaccuracy in the revised manuscript.

5. In line 235, “45.0% (15/60) succeeded” appeared. Is this correct?

We thank again the Reviewer for this comment and we corrected it in the revised manuscript.